# A Performance Evaluation of Vis/NIR Hyperspectral Imaging to Predict Curcumin Concentration in Fresh Turmeric Rhizomes

Michael B. Farrar [1,2,*], Helen M. Wallace [1,3], Peter Brooks [1,2], Catherine M. Yule [2], Iman Tahmasbian [4], Peter K. Dunn [2] and Shahla Hosseini Bai [1,3]

1   Genecology Research Centre, School of Science, Technology and Engineering, University of the Sunshine Coast, ML 40, Locked Bag 4, Maroochydore DC, QLD 4558, Australia; helen.wallace@griffith.edu.au (H.M.W.); pbrooks@usc.edu.au (P.B.); s.hosseini-bai@griffith.edu.au (S.H.B.)
2   School of Science, Technology and Engineering, University of the Sunshine Coast, ML 40, Locked Bag 4, Maroochydore DC, QLD 4558, Australia; pdunn2@usc.edu.au (P.K.D.); cyule@usc.edu.au (C.M.Y.)
3   Centre for Planetary Health and Food Security, School of Environment and Sciences, Griffith University, Nathan, Brisbane, QLD 4111, Australia
4   Department of Agriculture and Fisheries, Queensland Government, Toowoomba, QLD 4350, Australia; iman.tahmasbian@daf.qld.gov.au
*   Correspondence: mfarrar1@usc.edu.au

**Abstract:** Hyperspectral image (HSI) analysis has the potential to estimate organic compounds in plants and foods. Curcumin is an important compound used to treat a range of medical conditions. Therefore, a method to rapidly determine rhizomes with high curcumin content on-farm would be of significant advantage for farmers. Curcumin content of rhizomes varies within, and between varieties but current chemical analysis methods are expensive and time consuming. This study compared curcumin in three turmeric (*Curcuma longa*) varieties and examined the potential for laboratory-based HSI to rapidly predict curcumin using the visible–near infrared (400–1000 nm) spectrum. Hyperspectral images ($n$ = 152) of the fresh rhizome outer-skin and flesh were captured, using three local varieties (yellow, orange, and red). Distribution of curcuminoids and total curcumin was analysed. Partial least squares regression (PLSR) models were developed to predict total curcumin concentrations. Total curcumin and the proportion of three curcuminoids differed significantly among all varieties. Red turmeric had the highest total curcumin concentration (0.83 ± 0.21%) compared with orange (0.37 ± 0.12%) and yellow (0.02 ± 0.02%). PLSR models predicted curcumin using raw spectra of rhizome flesh and pooled data for all three varieties ($R^2_c$ = 0.83, $R^2_p$ = 0.55, ratio of prediction to deviation (RPD) = 1.51) and was slightly improved by using images of a single variety (orange) only ($R^2_c$ = 0.85, $R^2_p$ = 0.62, RPD = 1.65). However, prediction of curcumin using outer-skin of rhizomes was poor ($R^2_c$ = 0.64, $R^2_p$ = 0.37, RPD = 1.28). These models can discriminate between 'low' and 'high' values and so may be adapted into a two-level grading system. HSI has the potential to help identify turmeric rhizomes with high curcumin concentrations and allow for more efficient refinement into curcumin for medicinal purposes.

**Keywords:** curcumin; curcuminoids; hyperspectral imaging; jack-knifing; partial least squares regression (PLSR); turmeric (*Curcuma longa*); visible–near infrared (Vis/NIR)





## 1. Introduction

Hyperspectral imaging (HSI) is an emerging technology that has recently been used to non-destructively evaluate a variety of chemical compounds and quality indicators in soils and agricultural products (nuts, fruits, and vegetables) [1–4]. Traditional laboratory-based methods to detect compounds in plants are destructive and require specialised instrumentation and lengthy sample preparation procedures [5,6]. Additionally, soil physico-chemical properties, organic amendments, and crop growing conditions can lead to high variation in the chemical composition of plant materials [7,8]. Therefore, rapid and non-destructive HSI

methods are required to reduce cost and can improve efficiency in agricultural processes [9]. HSI can be adapted for different purposes including predicting nutrient concentrations in materials and identifying contamination and adulteration in foods and products refined from plants and animals [1,3,10].

Hyperspectral imaging utilises both spectral and spatial information collected by a single instrument and in doing so, generates large datasets [11]. As such, various methods exist to pre-process data and reduce dimensionality without losing important information. For example, many spectral wavelengths provide additional unnecessary information including light scatter and noise [1,12,13]. Transformations are applied to raw spectra to remove additive and multiplicative scatter effects prior to model development [14]. Minimal spectral transformation is preferred, with the aim, to produce robust models with high prediction accuracy. Transformations may be applied to data individually or in combination and many have specific calibration options leading to a multitude of user options that must be selected through trial and error [15]. For example, derivatives (particularly first and second) are useful for removing baseline shifts and selecting important wavelengths, whereas multiplicative scatter correction (MSC) can correct for effects such as light scattering due to variation in particle size [11]. Importantly, transformed spectra may not always perform better than untransformed spectra and choices must be made following analysis of any given data set [16]. Multivariate models are then developed with raw or transformed spectra using appropriate chemometric algorithms, such as partial least squares regression (PLSR) [11,16]. PLSR remains appropriate where collinearity exists between predictor variables and has successfully been used to analyze plant-based samples [17]. Hyperspectral imaging can be adapted to detect important compounds used in pharmaceutical products and in the plant materials they are derived from.

Curcumin is an important natural compound used for treatment of inflammatory disorders, carcinogenesis, and oxidative stress-induced pathogenesis [18–20]. Curcumin is refined from *Curcuma* spp. that contain varying levels of three polyphenolic curcuminoids: (1) curcumin, (2) desmethoxycurcumin, and (3) bisdemethoxycurcumin [21,22]. However, the genus *Curcuma* has 80 species and each species can have multiple varieties, for example, *C. longa* has 70 varieties in India [23]. It is important to know the level of curcumin in cultivated rhizomes and to identify high-yielding varieties. Traditional measurement of curcumin is carried out destructively by extracting curcumin from fresh rhizomes or dried powder and analysing it using high performance liquid chromatography (HPLC) or ultraviolet and visible spectrophotometry [7,24,25]. Isolation of curcumin from plant material is time consuming, laborious, and expensive, requiring specialised laboratory equipment and trained personnel [6,21]. Therefore, a rapid and non-destructive method to quantify curcumin in fresh rhizomes would represent a significant advantage for farmers and processors. Hyper- and multi-spectral detection of curcumin in turmeric powders using a variety of methods have been well investigated [26–28]. For example, HSI of midrange NIR spectra has successfully predicted curcumin concentration in turmeric powder [29]. However, prediction of curcumin using HSI images of fresh turmeric rhizomes has not yet been thoroughly explored.

Turmeric rhizomes contain an outer and inner zone with intermediate layers and individual vascular bundles, and as such, present difficulty for development of non-destructive technologies [22]. Therefore, studies describing the development of spectral methods to predict curcumin using rhizome skin, to the best of our knowledge, do not exist. Predicting chemical composition using plant material skin has always been challenging because spectral radiation needs to penetrate into the plant material to allow for internal measurement [30]. Additionally, many studies have explored hyperspectral methods using refined turmeric powders whereas studies using fresh rhizomes are limited. Therefore this study aimed to (1) compare total curcumin concentration and distribution of different curcuminoids in three varieties of *C. longa* grown in eastern Australia; and (2) evaluate the potential of PLSR models developed using visible–near infrared (Vis/NIR) spectra (400–1000 nm) to predict total curcumin concentration in fresh turmeric rhizomes. In

particular we explored the potential of hyperspectral imaging to predict curcumin non-destructively in fresh rhizome outer-skin and destructively using a cross-section of cut rhizome flesh. Three undescribed and commercially grown varieties of turmeric of different colours (yellow, orange, and red) were examined [31].

## 2. Materials and Methods

### 2.1. Experimental Design Overview

In this study, we used fresh turmeric rhizome samples to describe total curcumin concentration (%) and curcuminoid distribution in three undescribed local varieties and explore the potential for HSI and PLSR model development to predict total curcumin (%) using images of the fresh rhizome skin and flesh cross section (Figure 1).

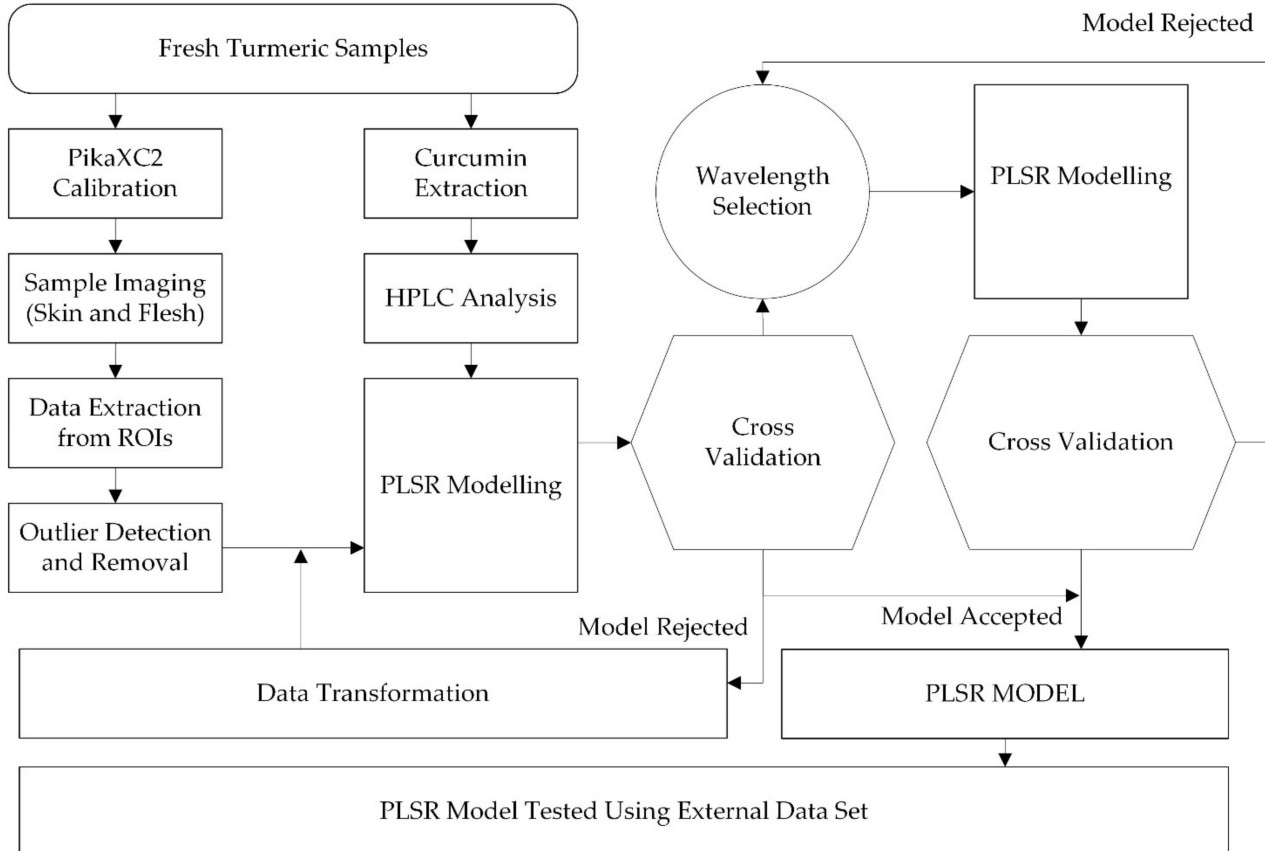

**Figure 1.** Flowchart illustrating the experimental design features and processes used to develop hyperspectral models for detection of curcumin if fresh turmeric rhizomes.

### 2.2. Sample Collection and Preparation

Three varieties (yellow, orange, and red) of turmeric (*C. longa)* were obtained from five sites at different organic commercial and hobby farms throughout South East Queensland and northern New South Wales (Figure 2). The sampling sites included two locations at a non-certified organic farm at Mount Mellum, (1) road paddock (26°47′38.77″S, 152°56′20.13″E) and (2) bottom paddock (26°47′37.17″S, 152°56′25.14″E), a certified organic farm at (3) Kandanga (26°22′35.86″S, 152°41′5.60″E) and (4) Terania Creek (28°59′25.70″S, 153°30′13.01″E) and a hobby farm at (5) Lake Macdonald (26°22′6.39″S, 152°55′55.09″E) (Table 1). Sampling locations were geographically disparate to capture variability in curcumin concentrations within the different varieties that may arise from soil, climate, and farm management practices. The orange variety is preferred for commercial cultivation due to rhizome size and ease of post-harvest processing [31].

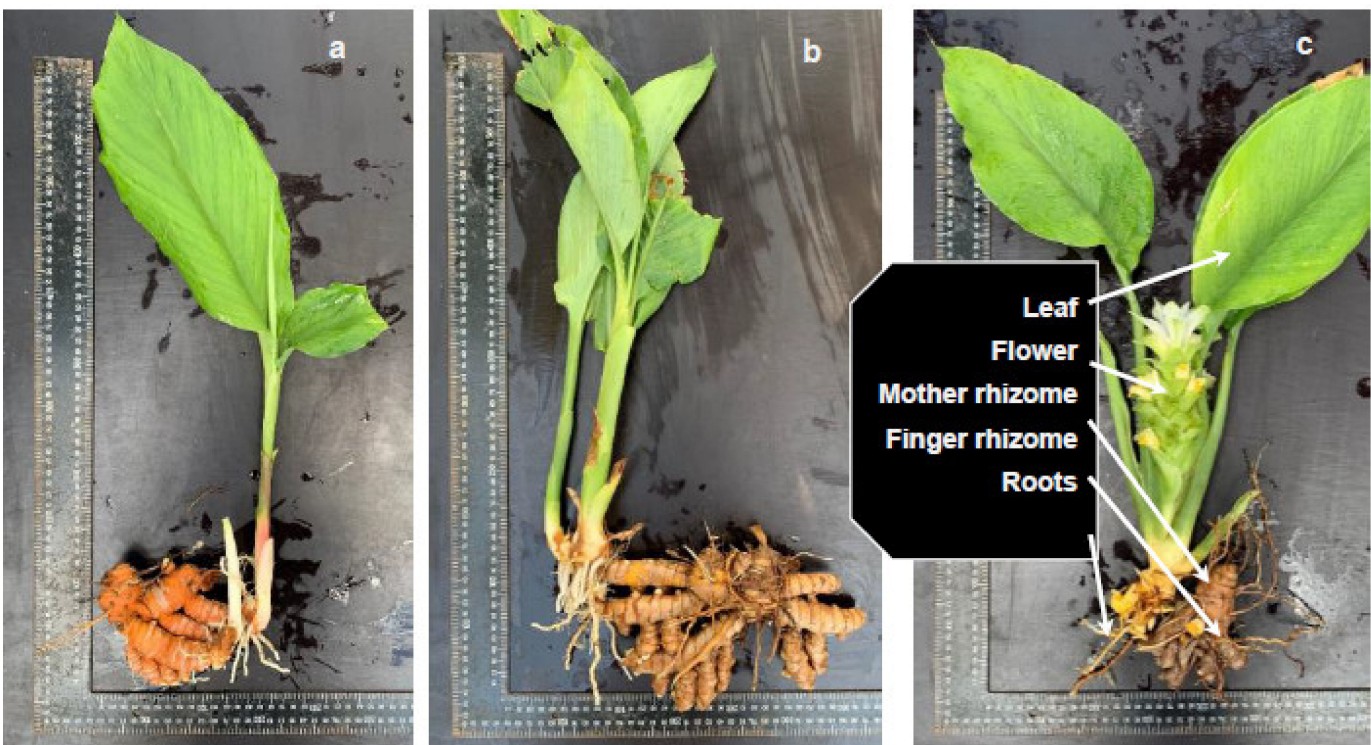

**Figure 2.** Rhizomes and foliage of the three turmeric (*Curcuma longa*) varieties orange (**a**), yellow (**b**), and red (**c**), examined in the study. Ruler shows scale in 1 mm increments and guide indicates turmeric anatomical features including: leaf, flower, mother rhizome, finger rhizomes, and roots.

**Table 1.** Distribution of sample variety and location of collection for all samples and in the calibration and external test sets.

| | All Samples | | | | Calibration Set | | | | Test Set | | | |
| | Variety | | | All Varieties | Variety | | | All Varieties | Variety | | | All Varieties |
| Location | Yellow | Orange | Red | | Yellow | Orange | Red | | Yellow | Orange | Red | |
|---|---|---|---|---|---|---|---|---|---|---|---|---|
| Mt. Mellum 1 | 6 | 55 | 7 | 68 | 6 | 44 | 5 | 55 | 0 | 11 | 2 | 13 |
| Mt. Mellum 2 | 9 | 17 | 8 | 34 | 8 | 13 | 7 | 28 | 1 | 4 | 1 | 6 |
| Lake MacDonald | 0 | 10 | 0 | 10 | 0 | 7 | 0 | 7 | 0 | 3 | 0 | 3 |
| Kandanga | 0 | 24 | 0 | 24 | 0 | 21 | 0 | 21 | 0 | 3 | 0 | 3 |
| Josh Rust | 4 | 12 | 0 | 16 | 2 | 8 | 0 | 10 | 2 | 4 | 0 | 6 |
| **Total** | 19 | 118 | 15 | 152 | 16 | 93 | 12 | 121 | 3 | 25 | 3 | 31 |

A total of 190 samples were collected from five study sites over a 12-month period between November 2018 and November 2019 to capture variation over the growth cycle. At the time of sampling, mother and connected finger rhizomes were lifted from the soil and one primary finger was cut from the mother rhizome, washed and airdried. Finger rhizomes were approximately 5–10 cm long. Samples from Terania Creek were express mailed to the laboratory to be delivered overnight. Samples were stored in a ziplock bags at room temperature for subsequent analysis within two days using the hyperspectral imaging system and HPLC.

### 2.3. Hyperspectral Imaging and Image Acquisition

Images were acquired using a laboratory-based visible–near infrared (Vis/NIR) hyperspectral imaging system (Figure 3a). The hyperspectral imaging system used incorporates a 12-bit push-broom line scanning camera with an operational spectral range of 400–1000 nm, a spectral sampling interval of ≈1.3 nm and spectral resolution of 2.3 nm (Resonon Pika XC2, Montana, USA) (Figure 3a). In the laboratory and immediately before hyperspectral imaging, a complete disc of each finger rhizome was cut laterally and approximately one-third from the base of the rhizome with a clean knife. The rhizome disc (cross-section) and

remaining rhizome was placed on a black mat 320 mm below the camera lens and scanned using the hyperspectral system (Figure 3).

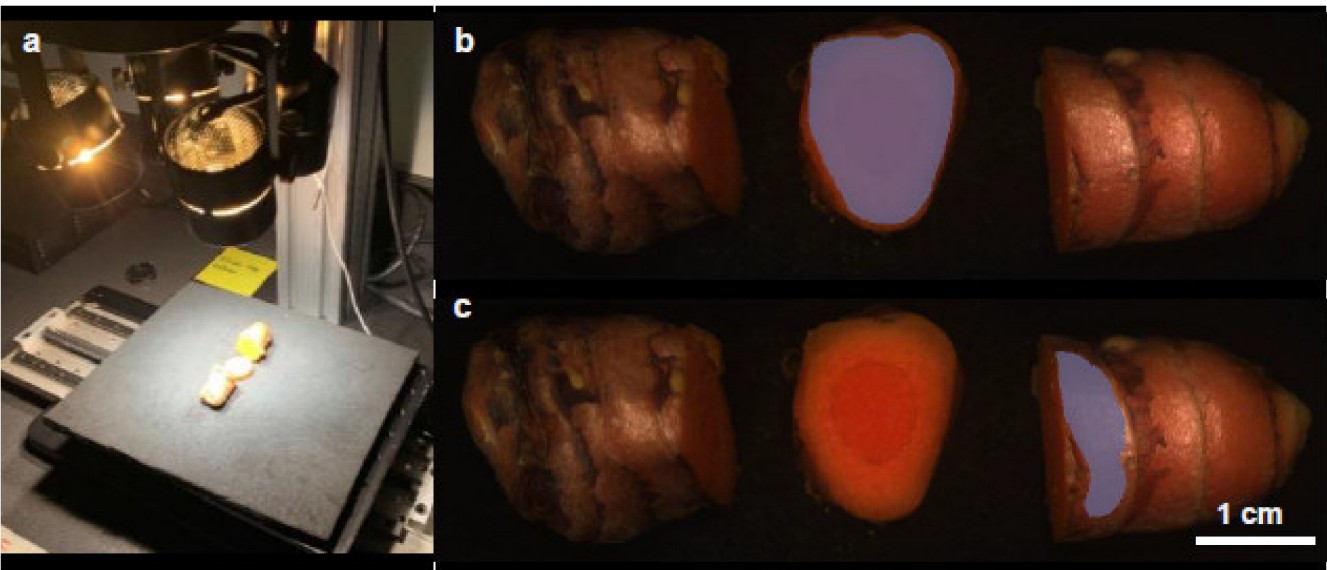

**Figure 3.** Scanning turmeric rhizome using hyperspectral imaging system Pika XC2 camera mounted 320 mm above the push-broom linear transition stage mat (**a**), selection of the rhizome flesh (cross-section) (**b**), and skin (**c**) region of interests (ROIs) used to extract mean spectral reflectance for each sample.

Finger rhizome samples were illuminated on a linear translation stage driven by electronic stepper motor for <1 min using four wide-spectrum (400–2500 nm), current controlled (12V DC) quartz halogen lights. Stage progression, data acquisition and image calibration were coordinated via SpectrononPro computer software v2.124 (Resonon Inc., Bozeman, MT, USA). Image calibration was undertaken prior to image acquisition in a darkened room and following calibration by removing dark current noise (dark calibration) and response correction (white calibration) by SpectrononPro software. Dark correction (D) was obtained by taking an image with the lens cap on. White correction (W) was obtained by taking an image of a highly reflective Lambertian material calibration sheet providing 99% reflectance. Reflectance was then calculated from raw spectral reflectance ($I_0$) using:

$$R = (I_0 - D)/(W - D), \tag{1}$$

Following scanning, the rhizome was reassembled to minimise oxidation and stored in a ziplock bag at room temperature until chemical extraction of curcumin within 6 h. Prevalence of curcumin cells varies between apical, nodal, and internodal regions therefore this method was employed to capture curcumin variation due to anatomical features [32]. Some of the rhizome discs sampled were internodal and others included a portion of undeveloped node (Figure 4).

*2.4. Data Pre-Processing: Region of Interest Selection, Outlier Detection and Data Set Assignment*

Regions of interest (ROI) were manually cropped from the background using the lasso tool function in SpectrononPro software by highlighting the (1) whole raw flesh inside the rhizome cross section, and (2) skin on the outside of the rhizome (Figure 3b,c). For each highlighted ROI, the average reflectance of selected pixels was extracted using SpectrononPro software v2.124 (Resonon Inc., Bozeman, MT, USA).

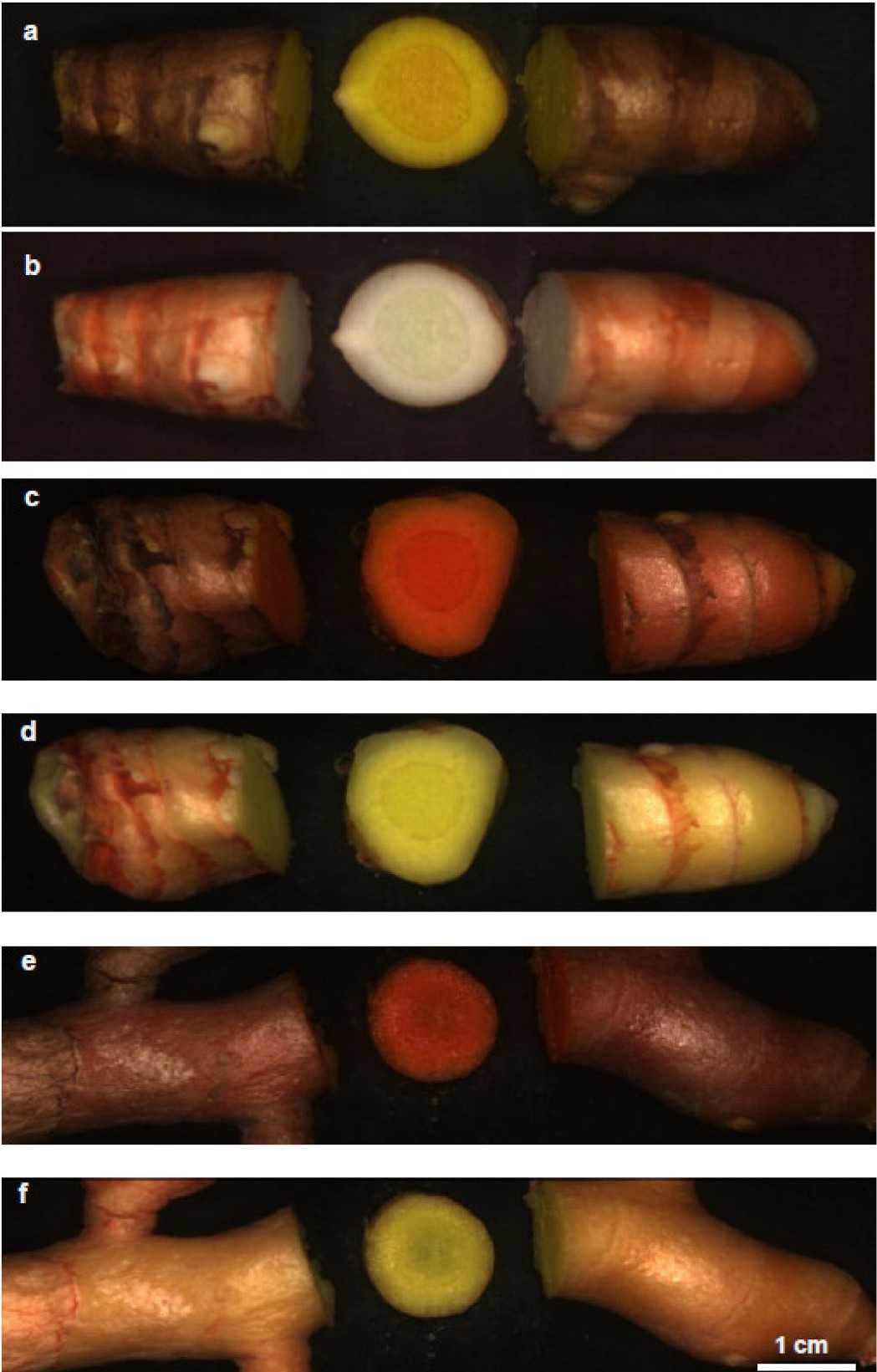

**Figure 4.** True colour and infrared images of the (**a**,**b**) yellow, (**c**,**d**) orange, and (**e**,**f**) red turmeric varieties examined in the study using hyperspectral images taken with PikaXC2. True colour images use spectral wavelengths: red = 640 nm, green = 550 nm, blue = 460 nm, and infra–red images use spectral wavelengths: red = 860 nm, green = 650 nm, blue = 550 nm. Scale bar represents 1 cm increment.

Outlying data points were detected and removed in order to establish a reliable and robust model [33]. Detection and removal of spectral outliers was carried out using principal component analysis (PCA) of the pooled varieties data set and visual inspection of the plotted spectra [34]. Three samples were spectrally and morphologically different and determined to be tuberous roots (*Yujin*), not rhizomes and were subsequently removed from the pooled data set [35]. Remaining samples (*n* = 152) were randomly assigned to calibration (*n* = 121) and test (*n* = 31) data sets (80/20%, respectively) for use in model development and evaluation (Table 1). This process was repeated until approximately relative proportions (80/20%) for each of the turmeric varieties were represented in both the calibration and test data sets (Table 1). Data sets were tested for centre and homogeneity of variance. Mean curcumin concentration and variation within the data in the calibration and test sets were not statistically different (*p* > 0.05) (Table 2). Samples of the orange variety (*n* = 118) only were used develop an additional model for a single variety only (Tables 1 and 2).

**Table 2.** Descriptive analysis of total curcumin (%) concentration for different varieties and locations in the calibration and test sets.

| | All Samples | | | | Calibration Set | | | | Test Set | | | |
|---|---|---|---|---|---|---|---|---|---|---|---|---|
| | Variety | | | All Varieties (%) | Variety | | | All Varieties (%) | Variety | | | All Varieties (%) |
| | Yellow (%) | Orange (%) | Red (%) | | Yellow (%) | Orange (%) | Red (%) | | Yellow (%) | Orange (%) | Red (%) | |
| Mean | 0.023 | 0.369 | 0.830 | 0.372 | 0.023 | 0.368 | 0.805 | 0.366 | 0.021 | 0.371 | 0.929 | 0.391 |
| SD | 0.022 | 0.121 | 0.224 | 0.227 | 0.023 | 0.120 | 0.232 | 0.226 | 0.018 | 0.126 | 0.192 | 0.241 |
| SE | 0.005 | 0.011 | 0.058 | 0.018 | 0.006 | 0.012 | 0.067 | 0.021 | 0.010 | 0.025 | 0.111 | 0.043 |
| CV | 0.957 | 0.328 | 0.270 | 0.611 | 0.997 | 0.327 | 0.288 | 0.618 | 0.838 | 0.340 | 0.207 | 0.617 |
| Min | 0.003 | 0.062 | 0.388 | 0.003 | 0.003 | 0.062 | 0.388 | 0.003 | 0.007 | 0.105 | 0.708 | 0.007 |
| Max | 0.076 | 0.673 | 1.133 | 1.133 | 0.076 | 0.673 | 1.133 | 1.133 | 0.041 | 0.648 | 1.050 | 1.050 |
| Skewness | 1.336 | −0.177 | −0.345 | 0.836 | 1.352 | −0.129 | −0.172 | 0.817 | 1.212 | −0.365 | −1.711 | 0.983 |

SD: Standard deviation; SE: Standard error; CV: Coefficient of variation. Not statistically different between calibration and test data sets at *p* < 0.05 independent student's *t*-test.

### 2.5. Curcumin Quantitation Using HPLC

The complete rhizome disc of flesh including skin (~1 g) was analysed for curcumin concentration. Samples were extracted with 50 mL Milli-Q $H_2O$ and blended for 2 min and made up to 100.0 mL with acetonitrile (ACN) in a 100.0 mL volumetric flask. Samples were sonicated and shaken until no remaining colour was visible in the turmeric tissue. Solutions were filtered through 0.45 µm filter prior to analysis. Chromasolv™ HPLC gradient grade ACN and HPLC grade Milli-Q water was used in all analyses.

The reverse-phase HPLC (Agilent Technologies 1290 Infinity II, quarternary pump, autosampler, column oven and diode array detector) used a 75 × 4.6 mm column (Phenomenex Synergi Fusion) with 4 µm particle size. Mobile phase A (MPA) was water:ACN (75:25%, *v*/*v*) and mobile phase B (MPB) was ACN (100%). The pump flow rate was 1.0 mL min$^{-1}$ and programmed to start at MPA/MPB 65:35 for 1.0 min, then grade to 45:55 at 6.0 min, then to 0/100 at 8.0 min, isocratic until 9.0 min, then back to 65:35 at 9.5 min and isocratic until 10.0 min. Injection was 20 µL, the column oven was set to 35 °C and detection was at 210 and 425 nm.

Standard aqueous solution of curcumin was prepared with curcumin (0.02 g) (Sigma-Aldrich, Saint Louis, MO, USA) sonicated and dissolved in 200 mL solvent (water 50%/ACN 50% (*v*/*v*)). Calibration standards were prepared from curcumin stock standard by serial dilution prior to analysis via HPLC as above. A calibration curve was constructed where the known concentration curcumin was plotted against the total peak area of curcumin (λ = 425 nm). Calibration curve for ten standards showed good linearity ($R^2$ = 0.9998) over the range of concentrations from 0.19 to 98.85% total curcumin. A further nine samples were analysed in triplicate to validate the HPLC method and are presented in the Supplementary Materials (Figure S1).

The three curcuminoids in extracts of rhizomes were separated well by HPLC (Figure 5). Curcuminoids: bisdemethoxycurcumin, desmethoxycurcumin, and curcumin eluted at

approximately 3.34, 3.59, and 3.84 min (detection 425 nm) respectively for all turmeric varieties analysed (Figure 5 and Figure S2). Total curcumin concentration was determined by linear regression using the standard curve. Therefore, the proportion of curcumin in individual samples was determined against the standard calibration curve by linear regression. Peak areas of the three separate curcuminoids were obtained and summed for use as total curcumin (%) reference values in further PLSR model development.

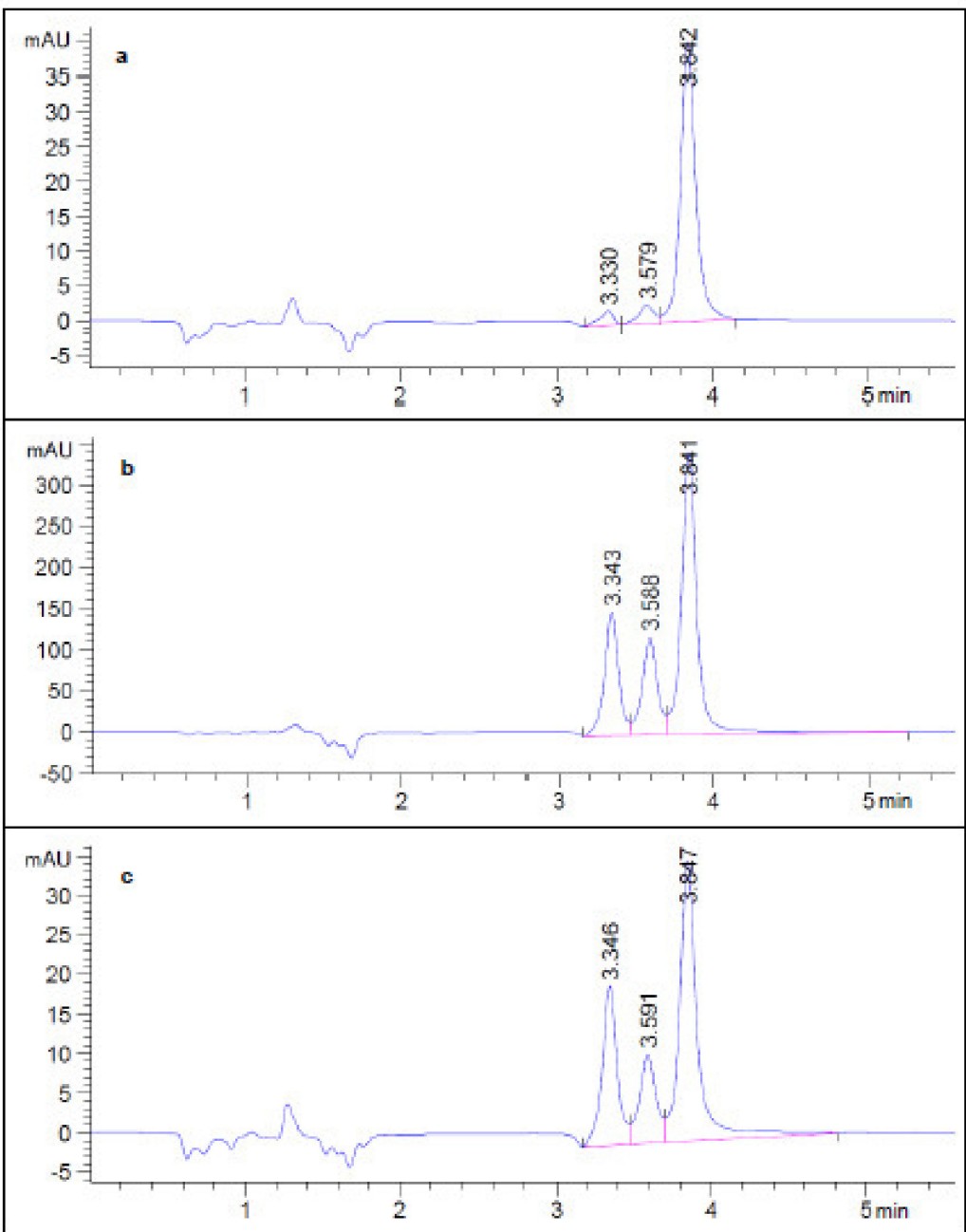

**Figure 5.** Typical HPLC chromatograms (λ = 425 nm) of the three turmeric (*Curcuma longa*) varieties yellow (**a**), orange (**b**), and red (**c**) used in the study. The chromatograms show three distinct peaks for the analogues of curcumin (bisdemethoxycurcumin, demethoxycurcumin and curcumin respectively) and different relative proportions between the different varieties.

## 2.6. Calibration Model Development and Spectral Data Transformations

LSR models were developed to describe the association between total curcumin concentrations in turmeric rhizome samples (described in an $n \times m$ matrix X) to their

relative reflectance in the full spectral range (described in a $n \times p$ matrix Y) [36,37]. The goal of PLSR is to make predictions of the matrix Y values using the information in matrix X when the number of predictors ($m$) is much larger than the number of observations $n$ (i.e., $m > n$). PLSR models are developed in the current study because the number of predictor variables ($m = 462$) is greater than the number of observations ($n \leq 121$) and therefore standard multivariate regression models are not appropriate [33]. Additionally, PLSR models are robust and reliable and can analyse noisy datasets and where collinearity (often a feature of hyperspectral data) exists among predictor variables [17,36]. PLSR finds a set of underlying variables (called *latent* variables, similar to Principal Components Analysis) common to X and Y that explain as much of the covariance between X and Y as possible [36,37]. The matrices X and Y are decomposed as $X = TP^T + E$ and $Y = UQ^T + F$, where these decompositions of X and Y take into account information from each other. The aim of the decomposition is to maximize the covariance between T (the latent variables from X) and U (the latent variables from Y). The PLSR model was fitted using the non-linear iterative partial least squares (NIPALS) algorithm [36].

Once the X and Y matrices have been decomposed and the latent variables described, the decomposition of X (the spectral information matrix) is then regressed on the dependent data Y (curcumin concentration matrix) such that:

$$\hat{Y} = TDQ^T = X\beta, \tag{2}$$

where $\beta$ is the regression coefficient matrix [36,37].

The optimal number of LVs for PLSR calibration models were selected at the minimum predicted residual error sum of squares (PRESS) of the validation set [38]. To avoid overfitting data and to obtain optimum performance from the model a full validation (leave-one-out) method was used [38,39]. This was done by systematically removing one sample from the data set and converging the model for the remaining samples. Then using the model to predict the sample left out and repeating the procedure for all samples [38].

Calibration models using both raw and transformed data were evaluated during model development using the optimal number of latent variables. PLSR models were developed using The Unscrambler v11 software package (CAMO Software Inc., Trondheim, Norway). A wide range of data transformation techniques were systematically performed on calibration spectra to scatter effects and increase signal to noise ratio [40]. Transformations explored included smoothing, Savitzky–Golay (first to forth) derivatives with varying polynomial orders, multiplicative scatter correction (MSC), extended multiplicative scatter correction (EMSC), orthogonal signal correction (OSC), standard normal variate (SNV), detrending and various combinations of techniques were investigated to select optimal calibration [40–42]. All computations for outlier detection and data transformations and were performed using The Unscrambler® software package (version 11, CAMO Software Inc., Trondheim, Norway).

Selection of the best-fitted calibration using the highest coefficient of determination for calibration ($R^2_c$) and cross-validation ($R^2_{cv}$) and the lowest root mean square error for calibration ($RMSE_c$) and cross-validation ($RMSE_{cv}$). The best models identified were further developed using wavelength selection.

*2.7. Wavelength Selection Using Jack-Knife Uncertainty Testing*

Hyperspectral data naturally contains very large numbers of variables (462 spectral bands in this study) and many variables have high collinearity which can decrease the ability of PLSR to successfully quantitate chemical reference values [38]. Therefore, selecting the most informative wavelengths can simplify and enhance interpretation and accuracy of models due to a reduction in latent variables in the models [43]. Normally, the analysis of variability of the PLSR regression coefficients gives information about the significance of variables. In this study, uncertainty testing (Jack-knifing) was used to select variables

by calculating the uncertainty estimates $s^2(b)$ associated with the regression coefficients ($b$) and loadings to remove unstable wavelengths in the models using:

$$s^2(b) = \frac{M-1}{M} \sum_{m=1}^{M} (b - b_m)^2, \tag{3}$$

where $M$ = the number of segments [44]. Using this method, wavelengths that are considered important in predicting curcumin well are retained in the model. This process was repeated until model accuracy ($R^2_c$ and $R^2_{cv}$) decreased from the previous model.

### 2.8. Evaluation of Calibration Models Using Test Data

The best fitted calibrations models were used to predict curcumin concentration in the test data. Accuracy of the calibration models to predict curcumin in new samples was evaluated by $R^2$ and RMSE for the test data set ($R^2_P$ and $RMSE_P$). Ratio of prediction to (standard error) deviation (RPD) was also calculated using [17,45]:

$$RPD = SD_Y / RMSE_p, \tag{4}$$

Using this indicator, RPD below 1.5 is considered 'very poor' and should not be used for prediction [46]. RPD between 1.5 and 2.5 can be considered 'useable' but with potential for increased efficacy, and 'excellent' if above 2.5 [46].

### 2.9. Statistical Analysis of the Different Turmeric Varieties

Descriptive analysis of the calibration and test data sets and tests for equality and difference between calibration and test sets, outlier detection, spectral data transformations and PLSR computations were performed using The Unscrambler® software package (version 11, CAMO Software Inc., Trondheim, Norway). The three varieties of turmeric were analysed for differences in total curcumin and the distribution of curcuminoids using one-way Analysis of Variance (ANOVA) and likelihood ratio test by generalized linear modelling using R (v4.0.0) in the RStudio (v1.2.5042) environment [47–49].

### 3. Results

### 3.1. Quantitation of Three Curcuminoids and Total Curcumin in Varieties of C. longa

Total curcumin (%) concentrations were significantly different among all varieties (Figure 6). The red variety had higher total curcumin (%) compared with the other varieties and ranged between 0.39 and 1.13% (Figure 6). The orange variety had higher curcumin (0.369%) compared with the yellow variety and ranged between 0.06 and 0.67% (Figure 6). The yellow variety had lower total curcumin compared with red and orange and ranged from <0.01 to 0.08% (Figure 6).

The turmeric varieties also contained different proportions of three curcuminoids (Figure 7). The red variety had higher proportion bisdemethoxycurcumin than both orange and yellow (Figure 7a). The orange variety contained higher proportion demethoxycurcumin compared to yellow and similar compared to red (Figure 7b) and the yellow variety had higher proportion curcumin compared with both orange and red (Figure 7c).

### 3.2. Descriptive Statistics for Data Sets Used in Model Calibration and Prediction

Mean curcumin concentration for the pooled data set (containing all three varieties) for calibration (0.366%) and test (0.391%) sets was not statistically different (Table 2). Mean curcumin concentration in the orange variety only data set for calibration (0.368%) and test (0.371%) sets was not statistically different (Table 2).

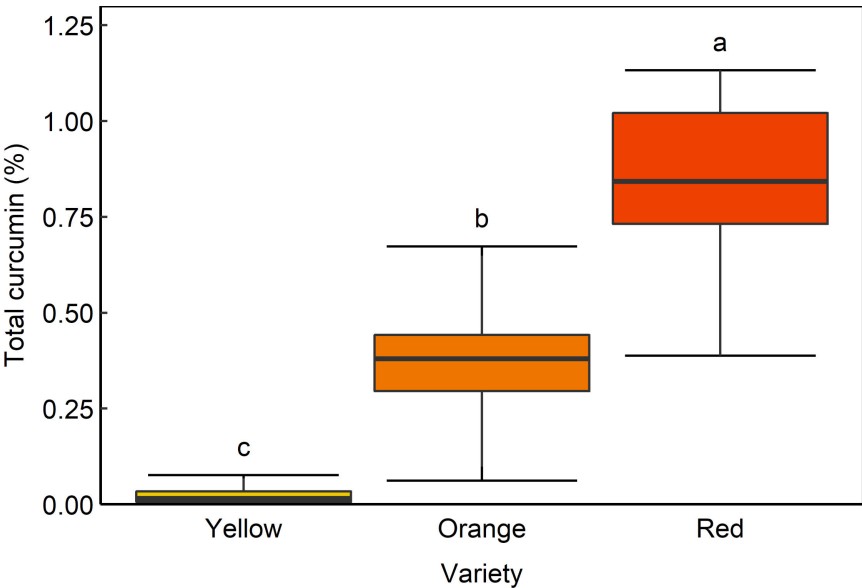

**Figure 6.** Total curcumin (%) concentration in fresh rhizomes of varieties (yellow, orange and red) (a, b, c) examined in the study.

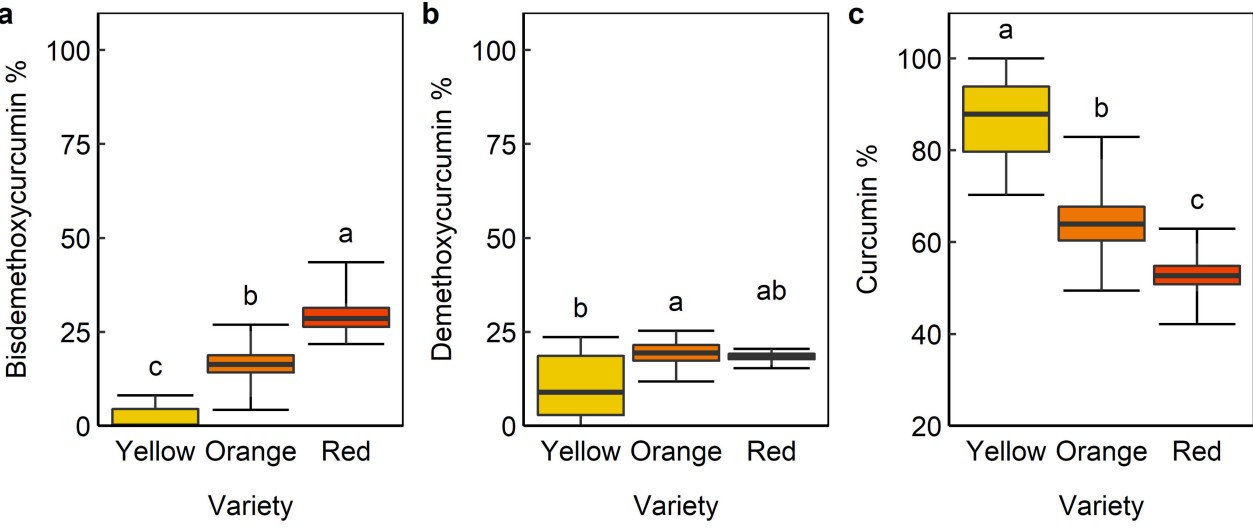

**Figure 7.** Percentage distribution of the three curcuminoids bisdemethoxycurcumin (**a**), demethoxycurcumin (**b**), and curcumin (**c**) within different varieties of turmeric examined in the study. Different letters within each panel are statistically different ($p < 0.001$).

### 3.3. Attributes of Developed Prediction Models Using the Full Spectrum and Transformed Spectra

Partial least squares regression (PLSR) models developed using images of rhizome flesh had higher accuracy and prediction performance than rhizome skin. Calibration models to predict total curcumin were first developed using pooled data using from three turmeric varieties and images of rhizome skin ($R^2_c = 0.64$, $RMSE_c = 0.136$) and flesh ($R^2_c = 0.83$, $RMSE_c = 0.093$) (Figure 8a,b). Models were examined for prediction performance using a test set of images of rhizome skin ($R^2_p = 0.37$, $RMSE_p = 0.188$, RPD = 1.28) and flesh ($R^2_p = 0.55$, $RMSE_p = 0.160$, RPD = 1.51) (Figure 8a,b). Additionally, reference values were compared with predicated values and deviation using samples in the test set (Figure 8c). A wide variety of transformations were investigated and models using transformed spectra with the highest $R^2_c$ are presented in Table 3. Application of Savitzky–Golay second order transformation with symmetric 11-point smoothing to

spectra increased $R^2_c$ for flesh ($R^2_c$ = 0.95, RMSE$_c$ = 0.050), however transforming spectra did not improve models using images of rhizome skin ($R^2_c$ = 0.63, RMSE$_c$ = 0.137) (Table 3).

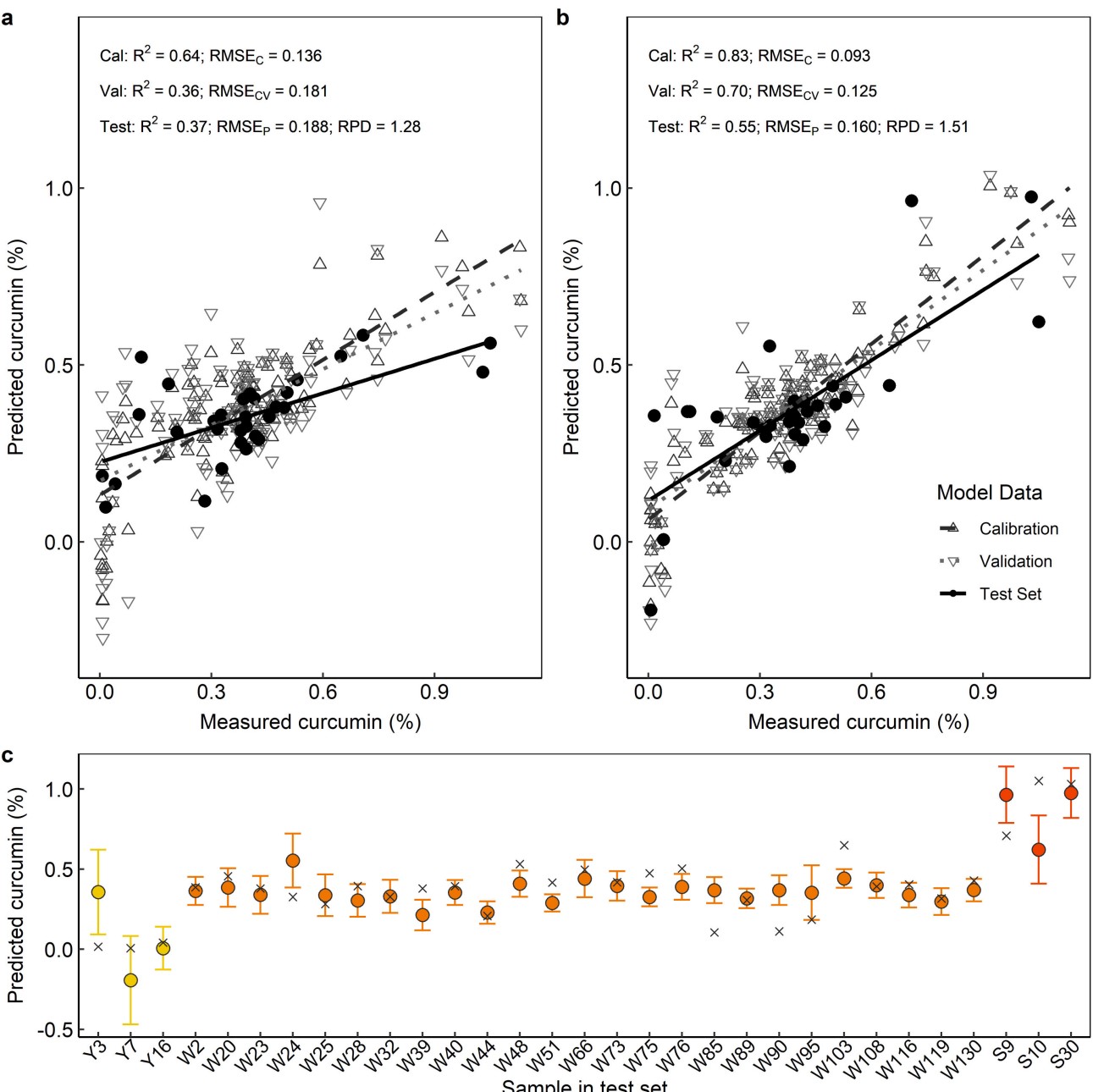

**Figure 8.** Models developed using raw reflectance spectra from three turmeric varieties (yellow, orange, and red) and all available wavelengths of rhizome skin (**a**), and rhizome flesh (**b**); figures show measured vs. predicted curcumin (%) concentration in the calibration set (Cal): open triangle; validation set (Val): open upside-down triangle; and test set: closed circle predicted curcumin concentration. Predicted curcumin (%) (closed circle) for individual samples in the test set using the flesh model plotted with measured reference value (cross) and deviation bars (similarity to calibration samples) for individual samples in the test data set (**c**).

**Table 3.** Performance of PLSR models to predict total curcumin (%) concentration of fresh turmeric rhizomes using a variety of spectral analytical techniques.

| Model | Images | Transformation | WL | LV | $R^2_C$ | $R^2$cv | $RMSE_C$ | $RMSE_{CV}$ | $R^2_P$ | $RMSE_P$ | RPD |
|---|---|---|---|---|---|---|---|---|---|---|---|
| **Three varieties pooled with:** | | | | | | | | | | | |
| wavelength selection | Flesh | | 95 | 8 | 0.71 | 0.64 | 0.12 | 0.14 | 0.55 | 0.16 | 1.52 |
| | Skin | | 9 | 4 | 0.25 | 0.18 | 0.19 | 0.21 | 0.27 | 0.20 | 1.19 |
| best transformation/s | Flesh | SG | 462 | 14 | 0.95 | 0.74 | 0.05 | 0.12 | 0.37 | 0.19 | 1.28 |
| | Skin | SG | 462 | 6 | 0.63 | 0.43 | 0.14 | 0.17 | 0.48 | 0.17 | 1.41 |
| best transformation/s + wavelength selection | Flesh | SG | 25 | 4 | 0.71 | 0.67 | 0.12 | 0.13 | 0.52 | 0.16 | 1.47 |
| | Skin | SG | 88 | 3 | 0.55 | 0.48 | 0.15 | 0.16 | 0.41 | 0.18 | 1.32 |
| **Orange variety only with:** | | | | | | | | | | | |
| wavelength selection | Flesh | | 28 | 8 | 0.70 | 0.58 | 0.07 | 0.08 | 0.51 | 0.09 | 1.46 |
| | Skin | | 180 | 2 | 0.20 | 0.17 | 0.11 | 0.11 | 0.11 | 0.11 | 1.15 |
| best transformation/s | Flesh | SG | 462 | 9 | 0.85 | 0.61 | 0.05 | 0.08 | 0.57 | 0.08 | 1.56 |
| | Skin | SG + MSC | 462 | 1 | 0.19 | 0.16 | 0.11 | 0.11 | 0.08 | 0.12 | 1.07 |
| best transformation/s + wavelength selection | Flesh | SG | 33 | 3 | 0.75 | 0.71 | 0.06 | 0.07 | 0.51 | 0.09 | 1.45 |
| | Skin | SG + MSC | 54 | 3 | 0.28 | 0.22 | 0.10 | 0.11 | 0.24 | 0.11 | 1.17 |

WL: Number of remaining wavelengths after wavelength selection to reduce dimensionality by uncertainty (Jack-knife) testing; LV: Optimal latent variables in the model; RPD: Ratio of prediction to deviation SG: Savitzky–Golay 2nd order derivative transformation with symmetric 11 point smoothing and 2nd order polynomial; SG + MSC: Savitzky-Golay 2nd order derivative transformation with symmetric 16 point smoothing and 2nd order polynomial followed by multiplicative scatter correction.

Calibration models were developed using data from the orange variety only from images of rhizome skin ($R^2_c$ = 0.21, $RMSE_c$ = 0.107) and flesh ($R^2_c$ = 0.85, $RMSE_c$ = 0.046) (Figure 9a,b). Models were examined for prediction performance using a test set of images of rhizome skin ($R^2_p$ = 0.21, $RMSE_p$ = 0.110, RPD = 1.15) and flesh ($R^2_p$ = 0.62, $RMSE_p$ = 0.076, RPD = 1.65) (Figure 9a,b). Additionally, reference values were compared with predicated values and deviation using samples in the test set (Figure 9c). Application of spectral transformations did not improve model accuracy for models using the orange variety only. Results of all models developed using the full spectrum and respective prediction performance using test data are depicted in Table 3. Models developed using pooled data from all three varieties were tested for prediction robustness using test data for rhizome skin ($R^2_p$ = 0.37, RPD = 1.28) and flesh ($R^2_p$ = 0.55, RPD = 1.51) (Table 3). Models using only orange variety data were tested for prediction robustness using test data for rhizome skin ($R^2_p$ = 0.21, RPD = 1.15) and flesh ($R^2_p$ = 0.62, RPD = 1.66) (Table 3). Full spectrum models were based on 462 spectral wavelengths (predictor variables). However, some wavelengths may be redundant, contain noise/interference and not be conducive to the development of multispectral systems suitable for real-time monitoring that require fewer variables.

### 3.4. Attributes of Developed Prediction Models Following Spectral Wavelength Selection

To improve the accuracy of models and reduce computer processing demands, the previous PLSR models were redeveloped following data reduction via wavelength selection. Models developed using reduced and transformed spectra using the pooled data set from images of the rhizome skin ($R^2_c$ = 0.55, $RMSE_c$ = 0.150) and flesh ($R^2_c$ = 0.71, $RMSE_c$ = 0.120) were improved compared to models using all raw spectral wavelengths (Table 3). Models were tested for prediction performance using test data for rhizome skin ($R^2_p$ = 0.41, RPD = 1.32) and flesh ($R^2_p$ = 0.52, RPD = 1.47) (Table 3).

Additional models were developed using data from the orange variety only. Reduced and transformed spectra of images of rhizome skin ($R^2_c$ = 0.28, $RMSE_c$ = 0.102) and flesh ($R^2_c$ = 0.75, $RMSE_c$ = 0.059) were also improved compared to using all raw spectral wavelengths (Table 3). Models developed using reduced and transformed spectra from images of only the orange variety were tested for prediction performance using test data for rhizome skin ($R^2_p$ = 0.24, RPD = 1.17) and flesh ($R^2_p$ = 0.51, RPD = 1.45). Results for all PLSR models developed following wavelength selection and respective prediction performance are reported in Table 3.

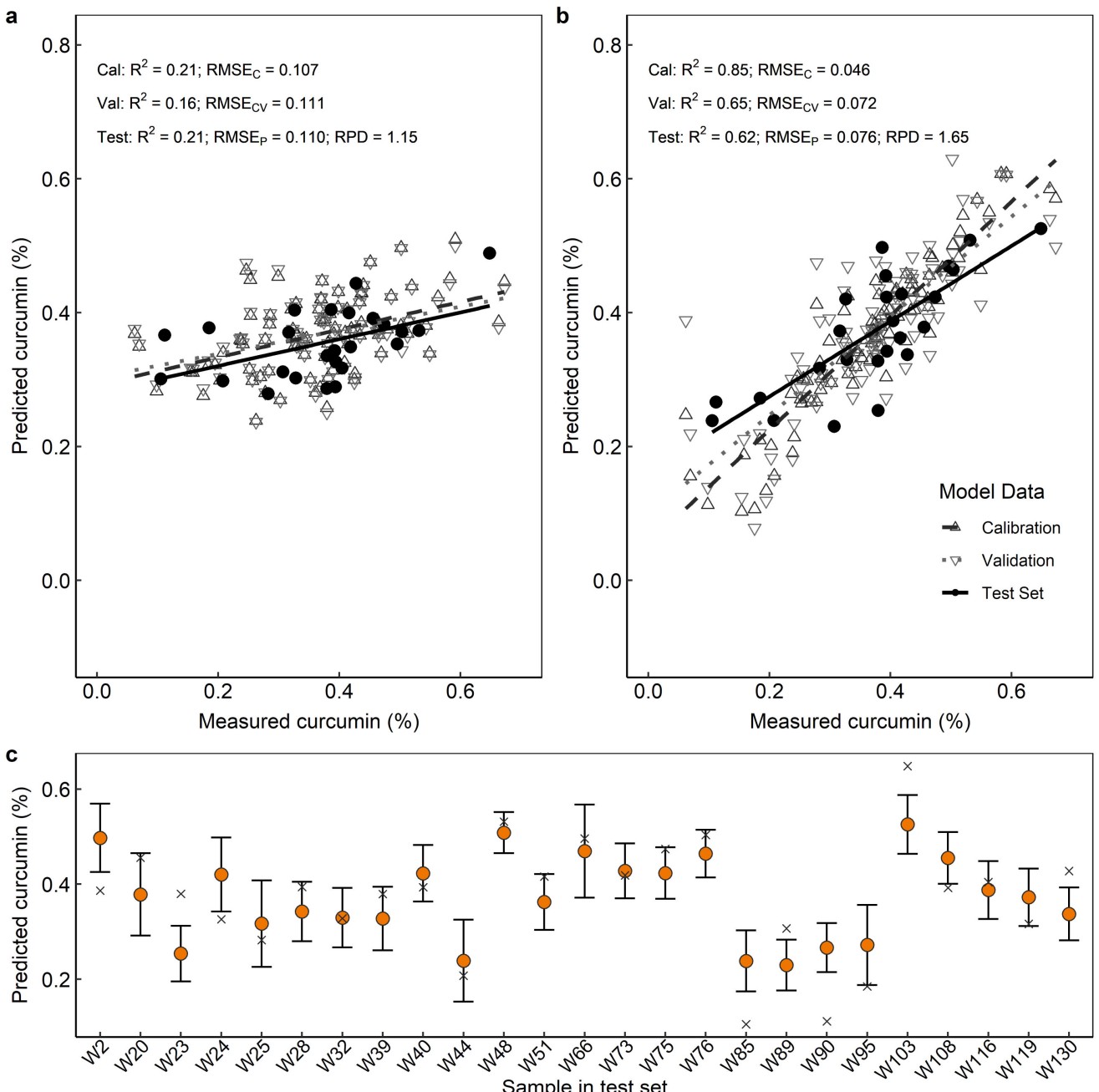

**Figure 9.** Models developed using raw reflectance spectra from orange variety turmeric only and all available wavelengths of rhizome skin (**a**), and rhizome flesh (**b**); figures show measured vs. predicted curcumin (%) concentration in the calibration set (Cal): open triangle; validation set (Val): open upside-down triangle; and test set: closed circle predicted curcumin concentration. Predicted curcumin (%) (closed circle) for individual samples in the test set using the flesh model plotted with measured reference value (cross) and deviation bars (similarity to calibration samples) for individual samples in the test data set (**c**).

## 4. Discussion

### 4.1. Curcumin Concentration among the Turmeric Varieties

The red variety had the highest total curcumin content varying between 0.39 and 1.34%. Variation in curcumin among varieties of *C. longa* and other species are well documented [50–52]. Curcumin content in rhizomes can vary with soil nutrient availability and agronomic conditions [51]. Distribution of curcuminoids for the red variety was similar to two identified varieties (Prathibha and Suguna) [52]. In this study, the red variety also contained proportionally more bisdemethoxycurcumin than the orange and

yellow varieties. Anatomical variation in the number and shape of curcumin cells has been associated with variability among rhizomes of *Curcuma* spp. [53]. Additionally, the number of curcumin cells are higher in the apical and nodal region compared with the internodal region [32]. Thus, high variation within samples of the red and orange variety could therefore be explained by the presence or absence of anatomical features of the individual samples such as apical regions or nodes.

*4.2. Assessment of Model Accuracy and Prediction Performance using Images of Rhizome Flesh*

Rhizome flesh using pooled data from all three varieties provided moderate curcumin prediction for calibration and validation ($R^2_c$ = 0.83, $R^2_{cv}$ = 0.70). The usefulness and accuracy of models developed in this study were evaluated on the basis of both calibration and validation model accuracy ($R^2_c$ and $R^2_{cv}$) and precision (RMSE$_c$, and RMSE$_{cv}$) followed by accuracy ($R^2_P$) and ratio of prediction to standard deviation (RPD) using the test data [54]. The respective coefficient of determination ($R^2$) values indicate the percentage variation in predicted curcumin concentration explained by the measured reference value. Therefore, higher $R^2$ values have improved prediction capability and specifically, models with an $R^2$ above 0.50 allow for prediction between nominally low and high concentration to be made and between and $R^2$ between 0.66 and 0.81 used for screening and approximate quantitative predictions [55]. Prediction accuracy ($R^2_p$ = 0.55) of curcumin prediction was low for the flesh pooled data model, despite high calibration accuracy ($R^2_c$ = 0.83). Therefore, this model would only be suitable to make predictions that discriminate between 'low' and 'high' values and as such may be suitable in a two-level grading system.

Transforming the data using a second order Savitzky–Golay derivative improved calibration accuracy ($R^2_c$ = 0.95) and was the most accurate of all calibration models developed. Higher $R^2$ values, between 0.92 and 0.96 reflect a model suitable for quality assurance applications [54,55]. However, prediction accuracy ($R^2_p$ = 0.37) further decreased compared with that of raw data and the model did not predict test data to an acceptable level (RPD = 1.28). The prediction performance (robustness) of a model is considered 'useable' if the ratio of performance to deviation (RPD) falls between 1.5 and 2.5 and 'excellent' if greater than 2.5 [46,54]. We suggest the decrease in prediction accuracy can be explained by 'data clusters' of samples with very low (<0.1%) curcumin in the calibration data combined with a lack of similar samples in the test data. The combined effect was reduced bias (regression slope) for test predictions resulting in poor prediction performance (Figure 8c). Therefore, we conclude that the best model, using data pooled from all three varieties, was developed using raw (un-transformed) spectra (RPD = 1.51) and may be suitable for adaption in a two-level grading system (Figure 8b).

We further hypothesised that large variation among the three varieties (low in yellow and high in red) coupled with under-representation of yellow and red samples contributed to reduced prediction accuracy in pooled data. Our hypothesis was then confirmed when the PLSR model developed using the orange variety only had higher $R^2_p$ compared with using data pooled from three varieties. This was the most accurate and robust model we developed and we consider it suitable for use in screening and approximate on-farm quantitation [46,55]. Additionally, models developed using one variety only were further improved by wavelength selection (Jack-knifing) making this method suitable for adaptation to smaller more portable multispectral imaging systems. We noted that not all samples in the test set were still predicted well (for example, W2, W23 and W103). Sample sizes > 200 have been successful for predicting quality parameters using images of flesh and skin in avocados [56]; therefore, we suggest increasing sample size prior to future model development. In general, each sample in the test set had similar deviation at the point of prediction in our study (Figure 9c). Deviation bars at the prediction point for a specific new sample in the test set, describe that sample's similarity to all samples in the calibration data [57]. Samples with small deviation are considered 'similar' to calibration samples and therefore represent 'reliable' predictions and samples with high deviation are considered 'dissimilar' to calibration samples and 'unreliable' predictions [57]. Therefore, confirming

test samples were similar to samples in the calibration data and that their prediction could be considered reliable.

We had initially hypothesised that pooling three varieties would provide a wider range of curcumin values leading to increased model accuracy and therefore better prediction performance. However, on the contrary, using images of the orange variety only increased prediction performance because of two reasons: orange only test samples were more similar to the sample in the calibration data and reduced clustering of the data, especially with very low curcumin concentrations. The pooled model often predicted test samples with relatively low (<0.01%) or high (>0.8%) curcumin poorly and was evidenced by large deviation (prediction reliability) at the prediction point. Using images from one turmeric variety only resulted in a calibration model that is more accurate and reliable at predicting new values. Our study suggested that models developed for curcumin prediction are variety dependent and we suggest that models should be developed for each variety prior to on-farm implementation.

### 4.3. Assessment of Model Accuracy and Prediction Performance using Images of Rhizome Outer-Skin

Our results indicated Vis/NIR hyperspectral imaging of rhizome skin did not predict curcumin in fresh turmeric well. The most accurate ($R^2_c$ = 0.63) model developed using skin images involved transformation (Savitzky–Golay second order derivative) of spectral data and was not improved by wavelength selection. Importantly, prediction accuracy ($R^2_p$ = 0.48) and prediction performance (RPD = 1.41) was considered 'below acceptable' and 'un-useable'. Interestingly, model redevelopment using images of the orange variety only did not improve accuracy nor prediction performance. This was in contradiction to the models we developed using images of flesh. Internal prediction of parameters using images of the outer-skin of intact samples are always challenging because of heterogeneity and accuracy can be increased by milling, mincing, or grinding [15]. Internal curcumin prediction may have been possible using images of rhizome skin if light penetrated deeper through the skin and into the flesh or had we measured curcumin in the skin only which could then be correlated to curcumin concentration measured in the flesh. We did not measure light penetration in our samples. However, shorter wavelengths penetrate deeper into biological samples than longer ones [58]. Additionally, image deep learning in other studies has shown to improve prediction accuracy which can be further explored for skin images [59]. Therefore, we suggest further investigation using different detectors and spectral wavelengths is still required.

### 4.4. Implications of This Scoping Study

A costly and laborious initial process is required to develop PLSR models using HSI that require sample reference values to first be attained using traditional laboratory methods. However, the initial cost of HSI establishment including a benchtop HSI system, may not differ significantly compared with the cost of a HPLC instrument (Figure S3). After HSI modelling has been established, the cost to analyze a new sample can be reduced by up to 76% and processing time can be decreased by 66% per sample (30 to 10 min) when compared with HPLC. However, one important compromise is lower accuracy of curcumin prediction when using HSI rather than by HPLC. However, developments in machine and deep learning of HSI have recently increased the prediction accuracy of plant components [59]. Machine learning can be further explored to increase accuracy of curcumin prediction [59]. Additionally, benefits of higher accuracy using traditional methods is offset by the ability of HSI methods to analyze large quantities of samples due to reduced cost. Therefore, HSI would provide information across larger samples sizes and within short-time frames, such as in post-harvest processing and quality assurance [60]. This technology has the potential to be upscaled and used on farms following engineering design and modification to suit post-harvest processing lines and most recently has successfully been incorporated into commercial sorting of potato for sugar-end defect [61]. Adaption of HSI to help with grading rhizomes would be particularly useful where fresh turmeric rhi-

zomes are being harvested for end use refinement into curcumin-based medicinal and/or pharmaceutical products.

## 5. Conclusions

The red variety of *C. longa* contained the highest curcumin and we recommend farmers cultivate this variety where curcumin yield is desired. The results from this study demonstrate that Vis/NIR hyperspectral imaging combined with PLSR has potential to predict curcumin in fresh turmeric using images of rhizome flesh but not outer-skin. During harvesting and washing, finger rhizomes are often broken from the mother rhizome and are still marketable, therefore, scanning of any broken rhizome pieces, randomly selected from a process batch, and using the PLSR models we developed may allow for on-farm means-based grading of packaged rhizomes under a two-level system. Developing models for each variety (rather than pooling varieties) improved prediction performance and reliability and is a more appropriate approach than using pooled data. Models developed using one turmeric variety (orange) were more accurate and had higher prediction performance and reliability. These were further improved by wavelength selection (Jack-knifing) making this method suitable for adaptation to smaller more portable multispectral imaging systems. However, larger sample sizes for each specific variety and investigation of data collected from other spectral regions should be undertaken in future studies. Additionally, this method should be examined to predict individual curcuminoids and emerging image deep learning algorithms may further improve model prediction performance in the future.

**Supplementary Materials:** The following are available online at https://www.mdpi.com/article/10.3390/rs13091807/s1, Figure S1: Linear regression of total curcumin (%) concentration extracted and analysed by UV-Vis and HPLC (**a**) and boxplot showing total curcumin (%) concentration of triplicate analysis conducted for $n = 3$ of each variety of yellow (Y), orange (W) and red (S) turmeric rhizomes (**b**). To validate the HPLC method triplicate replications using three samples of each variety were analysed by both HPLC and UV-Vis to correlate results. Samples were prepared and analysed using the HPLC method above and Genesis 20 UV-Vis spectrometer in Shimadzu quartz crystal cuvettes with 10 μm light path at 425 nm (Thermo Scientific, Waltham, Massachusetts, USA). Curcumin concentration values for both instruments were correlated with $R^2 = 0.999994$ (Figure S1a). Confirming accuracy of the HPLC method. Curcumin (%) concentration for the triplicate samples in the HPLC method validation set are presented in (Figure S1b), Figure S2: Spectral peaks used to identify the curcuminoids at retention time bisdemethoxycurcumin at 3.349 min (**a**), demethoxycurcumin at 3.3593 min (**b**), and curcumin at 3.844 min (**c**), Figure S3: Cost-benefit analysis comparing traditional laboratory based HPLC curcumin detection and prediction using hyperspectral imaging and PLSR methods.

**Author Contributions:** Conceptualization, M.B.F., P.B. and S.H.B.; Data curation, M.B.F.; Formal analysis, M.B.F., P.B. and I.T.; Funding acquisition, M.B.F., H.M.W. and S.H.B.; Investigation, M.B.F.; Methodology, M.B.F., P.B. and S.H.B.; Project administration, M.B.F.; Resources, M.B.F. and H.M.W.; Software, M.B.F., I.T., P.K.D. and S.H.B.; Validation, M.B.F., I.T., P.K.D. and S.H.B.; Visualization, M.B.F.; Writing—original draft, M.B.F. and S.H.B.; Writing—review and editing, M.B.F., H.M.W., P.B., C.M.Y., I.T., P.K.D. and S.H.B. All authors have read and agreed to the published version of the manuscript.

**Funding:** M.B.F. was funded by University of the Sunshine Coast and received an Australian Government Research Training Program (RTP) Scholarship to undertake this research. M.B.F. received support from Mt. Mellum Horticulture via donation of turmeric rhizomes, nursery space, and facility support including irrigation infrastructure and water.

**Data Availability Statement:** The datasets used and/or analysed during the current study are available from the corresponding author on reasonable request.

**Acknowledgments:** Michael B Farrar was supported by University of the Sunshine Coast and received an Australian Government Research Training Program (RTP) Scholarship to undertake this research. The authors would like to acknowledge Jess and Jesse Joyce at Little Bunya Farm, Josh Rust at Nightcap Functional Foods, Mellum Horticulture, and Martyn Elliot for farm access and providing

turmeric samples. Simon Williams, Linda Pappalardo, and Michael Nielson for help with laboratory analysis. The authors also thank Michael Joyce at Mighty Bean Foods for initialising a curiosity in local turmeric varieties and allowing access to initial planting rhizomes from 'Wally'.

**Conflicts of Interest:** M.B.F. is a part owner of Mt Mellum Horticulture where the study took place. No financial gain will be received through publication of this manuscript. M.B.F. did not influence the results and all data and interpretations have been independently assessed. The funders had no role in the design of the study; in the collection, analyses, or interpretation of data; in the writing of the manuscript, or in the decision to publish the results.

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
