# Peer review of "A Performance Evaluation of Vis/NIR Hyperspectral Imaging to Predict Curcumin Concentration in Fresh Turmeric Rhizomes"

_remotesensing, doi:10.3390/rs13091807_

Round 1
Reviewer 1 Report
Dear authors,
I don't have any concerns regarding this article.
Personally I would like less description in the name of each Figure, but I will leave this for editor to decide.
Figure 4 - if you could slightly move numbers on graph, for better visibility.
Eq. 3 - is written in different form than others. Please check.
Author Response
Please see the attachment to find our responses in the word document.

Reviewer 2 Report
The article is very interesting and potentially interesting to scientists dealing with similar topics. It is well designed and possibly should be cut in half pictures 4 and 5.
Author Response

(The authors gave the same response as above.)

Reviewer 3 Report
The manuscript by Farrar et al. presents results from a study predicting curcumin concentration in fresh turmeric rhizomes using hyperspectral images. Though the information provided in this manuscript is interesting, serious revisions of the introduction section and some statements throughout the manuscript is necessary to ensure that the justification of the study is aligned with the study design.
The major limitation of this manuscript is the alignment between the justifications of the study and the study design.
- Lines 43-46: “Traditional laboratory-based methods to detect compounds in plants are destructive and require specialized instrumentation and precision sample preparation, and are therefore, laborious, time-consuming and expensive”. The proposed method in this manuscript also requires “cut-flesh”, which is destructive. What is the difference between the traditional methods and the proposed method in terms of destructive vs. non-destructive?
- Lines 43-46 also imply that the traditional methods are more time-consuming and expensive than HSI or something similar to the proposed method. You will need to include a cost-benefit analysis (e.g., differences in time, $, and accuracy between HSI based methods and the traditional methods) to support this statement or remove it from the introduction. The statement was based on assumption and expectation however better to be supported with facts along with data analysis.
- Approximately 30 varieties of turmeric have been recognized by biologists and I believe that the three commercially varieties of turmeric have been well studied by biologists and farmers would know the different values of these three varieties before they decided to grow them on their farms. From my perspective it is impractical: “scanning of any broken pieces and using the models we developed may allow for on-farm grading of rhizomes” (Lines 476-477 in the conclusion section).
- As mentioned in the introduction, HSI of midrange NIR spectra has been proven valid to predict curcumin concentration in turmeric power but prediction of curcumin using HSI in the Vis/NIR of 400-100nm has not been well studied. Is it already a good justification for the study?
I recommend its publication after a major revision of the introduction section.
Author Response
Please see the attachment for our responses in the word document.

Reviewer 4 Report
General Comments:
Overall, this paper compared curcumin in three turmeric varieties and examined the potential for laboratory-based HIS to predict curcumin using the visible-near infrared spectrum. The authors analyzed the distribution of curcuminoids and total curcumin; and developed the Partial least squares regression (PLSR) models to predict total curcumin concentrations. The manuscript is logically organized and clearly presented. The theme of this paper is clear, and the authors have done a nice job in analyzing the research results. Besides, there is enough validation to explain the credibility of the model. However, the content of this paper needs to be adjusted. Some suggestions that might help authors improve the paper are as follows:
Specific Comments:
- The manuscript lacks sufficient discussion/analysis on the proposed method. What is the innovation of the proposed method? What could readers learn from the proposed approach? What could be the motivation for the readers to apply the proposed approach to other data sets/studies?
- There is not enough literature review in this manuscript. After the literature review, what are their gaps, and what is the motivation for authors to propose this method?
- The resolution of some figures is poor. The figures should be consistent with the format required by the journal.
- In terms of the experimental methodology, the authors could provide a figure to describe the experiment’s design and process, and make a more detailed description about why choose these methods.
Author Response

(The authors gave the same response as above.)

Reviewer 5 Report
I have seen usage of aircraft hyperspectral imaging system for determination of foliar chlorophyll, so I do not see problem why it would be not used for measuring of curcumin, althrough the estimation is from roots.
The hyperspectral imaging system used has an 126 operational spectral range of 400-1000 nm at 1.3 nm resolution (resulting in 462 spectral 127 bands) - are you sure you are understanding good difference between resolution (in mean FWHM) and step of data acquisition what I would understand to be 1.3 if there are 460 bands between 400 and 1000 nm. I would suggest resolution (FWHM) would be lower, I mean broader spectral bands are measured.
Line 137 - current controlled quartz halogen lights – what was the spectrum of the illuminating light because it may determine and highly influence spectrum measured.
Line 193 - programmed to started at ?
Line 299 - red variety had higher cur-299 cumin compared with both yellow and orange (Figure 6c) - I see something different in figure 6c and it repeats in first sentence of results and in conclusion, is it all right?
Author Response

(The authors gave the same response as above.)

Round 2
Reviewer 3 Report
Thanks for the revision. I am happy with the changes.
Author Response
Thank you for reviewing and accepting our major revisions.